# Rice Bran and Vitamin B6 Suppress Pathological Neovascularization in a Murine Model of Age-Related Macular Degeneration as Novel HIF Inhibitors

**DOI:** 10.3390/ijms21238940

**Published:** 2020-11-25

**Authors:** Mari Ibuki, Deokho Lee, Ari Shinojima, Yukihiro Miwa, Kazuo Tsubota, Toshihide Kurihara

**Affiliations:** 1Laboratory of Photobiology, Keio University School of Medicine, Tokyo 160-8582, Japan; shirayuki0727@yahoo.co.jp (M.I.); deokholee@keio.jp (D.L.); ari.shinojima@keio.jp (A.S.); yukihiro226@gmail.com (Y.M.); 2Department of Ophthalmology, Keio University School of Medicine, Tokyo 160-8582, Japan; 3Animal Eye Care•Tokyo Animal Eye Clinic, Tokyo 158-0093, Japan; 4Tsubota Laboratory, Inc., Tokyo 160-0016, Japan

**Keywords:** hypoxia-inducible factor, age-related macular degeneration, vascular endothelial growth factor, food ingredients, rice bran, vitamin B6, retinal pigment epithelium

## Abstract

Pathological neovascularization in the eye is a leading cause of blindness in all age groups from retinopathy of prematurity (ROP) in children to age-related macular degeneration (AMD) in the elderly. Inhibiting neovascularization via antivascular endothelial growth factor (VEGF) drugs has been used for the effective treatment. However, anti-VEGF therapies may cause development of chorioretinal atrophy as they affect a physiological amount of VEGF essential for retinal homeostasis. Furthermore, anti-VEGF therapies are still ineffective in some cases, especially in patients with AMD. Hypoxia-inducible factor (HIF) is a strong regulator of VEGF induction under hypoxic and other stress conditions. Our previous reports have indicated that HIF is associated with pathological retinal neovascularization in murine models of ROP and AMD, and HIF inhibition suppresses neovascularization by reducing an abnormal increase in VEGF expression. Along with this, we attempted to find novel effective HIF inhibitors from natural foods of our daily lives. Food ingredients were screened for prospective HIF inhibitors in ocular cell lines of 661W and ARPE-19, and a murine AMD model was utilized for examining suppressive effects of the ingredients on retinal neovascularization. As a result, rice bran and its component, vitamin B6 showed inhibitory effects on HIF activation and suppressed *VEGF* mRNA induction under a CoCl_2_-induced pseudo-hypoxic condition. Dietary supplement of these significantly suppressed retinal neovascularization in the AMD model. These data suggest that rice bran could have promising therapeutic values in the management of pathological ocular neovascularization.

## 1. Introduction

Ocular pathological neovascularization is a leading cause of blindness worldwide [1,2]. It affects our lives in all age groups from children to elderly with various names of diseases such as retinopathy of prematurity (ROP), diabetic retinopathy (DR), retinal vein occlusion and age-related macular degeneration (AMD) [3]. To date, a blockade of vascular endothelial growth factor (VEGF) using anti-VEGF antibodies has been applied for the treatment of these diseases [4], as VEGF plays a crucial pathological role in the development of these diseases [5,6]. Even though small concentrations for the treatment to a local target site may not affect systemic side effects of the drugs, repetitive administrations of anti-VEGF drugs for chronic therapies could alter a systemic or local VEGF amount which may be required for normal vascular and neuronal maintenance [4]. Moreover, anti-VEGF drugs are still ineffective in some cases, especially in patients with AMD [7].

Hypoxia-inducible factor (HIF) plays a strong transcription regulator of VEGF induction under hypoxic and other stress conditions [8]. Under hypoxic conditions, HIF translocates to the nucleus and binds the hypoxia response element (HRE), inducing hypoxia responsive gene expressions including VEGF as well as B-cell lymphoma 2 interacting protein 3 (BNIP3) and phosphoinositide-dependent kinase 1 (PDK1) [9,10]. As the HIF/VEGF axis is a strong pathological pathway for neovascularization [11,12], inhibiting HIF activation could be an attractive target for antineovascularization therapies. Moreover, HIF expression was observed in human choroidal neovascular membranes in patients with AMD [13,14], and retinal pigment epithelial (RPE) cells resided in those membranes were localized with the presence of HIF and VEGF [14].

Previously, we demonstrated HIF inhibition-suppressed retinal neovascularization genetically and pharmacologically in murine models of oxygen-induced retinopathy (OIR), one of the ROP models, and laser-induced choroidal neovascularization (CNV), one of the AMD models [15,16,17,18,19]. Moreover, another HIF inhibitor, halofuginone, a synthetic derivative of febrifugine isolated from hydrangea, exerted retinal protection in a murine ischemia–reperfusion model [20].

In our daily lives, we consume extensive amounts of food. Even though food has been classically perceived as a simple means for energy production and body construction, their potentials have expanded to drug discovery and development for various diseases [21,22,23]. Diet that contains omega-3 fatty acids could enhance cognitive functions in humans [24]. A suggestive association of a vegetable-rich and low carbohydrate diet and a lower risk of early paracentral visual loss in primary open-angle glaucoma was recently reported in data from three United States cohorts [25]. There are still lots of food ingredients that could be unraveled in terms of potent therapeutic effects that they possess on various diseases including retinal neovascularization.

Along with this, we obtained various types of food ingredients and attempted to find novel effects of them through drug screenings: inhibitors of HIF activation. After screenings, we investigated therapeutic effects of positively selected ingredients on retinal neovascularization in a murine model of laser-induced CNV.

## 2. Materials and Methods

### 2.1. Animal

All experimental procedures were approved by the Ethics Committee on Animal Research of the Keio University School of Medicine (approved number #16017-2 on 12 October 2018) and followed with the ARVO Statement for the Use of Animals in Ophthalmic and Vision Research and the international standards of animal care and use in ARRIVE (Animal Research: Reporting in Vivo Experiments) guidelines (http://www.nc3rs.org.uk/arrive-guidelines). C57BL/6 and BALB/c male mice were obtained from CLEA Japan (Tokyo, Japan) and raised in a standardized temperature-controlled environment under a 12 h light-dark cycle with free access to water and food.

### 2.2. Cell Culture

A mouse cone photoreceptor 661W cell line was maintained in DMEM (Cat #08456-36, Nacalai Tesque, Kyoto, Japan) media with 10% FBS and 1% streptomycin-penicillin under an atmospheric condition containing 5% CO_2_ at 37 °C. A human cell line for retinal epithelial ARPE-19 was maintained in DMEM/F-12 (Cat #C11330500BT, Gibco, Waltham, MA, USA) media with the same supplements above in the same atmospheric condition.

### 2.3. Food Ingredient Preparation and Luciferase Assay

Food ingredients for the screening were prepared as listed in Table A1. Then, a luciferase assay for drug screenings on inhibition of HIF activation was performed as previously described in our papers [15,16,17,20]. Briefly, 661W and ARPE-19 cells, transfected with a HIF-luciferase reporter gene construct (Cignal Lenti HIF Reporter, Qiagen, Venlo, Netherlands) encoding a firefly luciferase gene under a control of the HRE, were seeded and treated with a well-known HIF inducer CoCl_2_ (200 µM, cobalt (II) chloride hexahydrate, Wako, Osaka, Japan). To evaluate inhibitory effects of food ingredients on HIF activation, cells were cotreated with 1 mg/mL of each ingredient and CoCl_2_. 24 h after incubation, luminescence was measured using Dual-Luciferase Reporter Assay System (Promega, Madison, WI, USA).

### 2.4. MTT Assay

For determination of cytotoxicity of food ingredients under a CoCl_2_-induced hypoxic condition, ARPE-19 cells were seeded in 96-well plates and the ingredients were treated to the cells for 12 h. A total of 10 µL of MTT solution (Cat #M2128, Sigma, St. Louis, MO, USA) was added to each well and incubated for 2 h at 37 °C. After labeling the cells with MTT, media were removed from the wells and 100 µL of DMSO was added to each well. Then, absorbance of colored solution in the wells was measured at 540 nm (Synergy HT Multi-Mode Microplate Reader, Winooski, VT, USA).

### 2.5. Quantitative PCR and Western Blotting

RNA extraction, cDNA synthesis and real-time quantitative PCR were performed as same as previously described in our papers [15,16,17,20]. Primers used are listed in Table 1. Fold changes between levels of different transcripts were calculated by the ΔΔC*_T_* method.

Protein extraction, electrophoresis and visualization of protein bands were performed as the same as previously described in our papers [15,16,17,20]. Antibodies used were anti-HIF-1α (1:1000, Cat #36169, Cell Signaling Technology, Danvers, MA, USA), anti-HIF-2α (1:1000, Cat #NB100-122, Novus Biologicals, Centennial, CO, USA) and anti-β-Actin (1:5000, #3700, Cell Signaling Technology, Danvers, MA, USA). For visualization, HRP-conjugated secondary antibodies (1:5000, GE Healthcare, Chicago, IL, USA) were used. Blotting was quantified using NIH ImageJ software (National Institutes of Health, Bethesda, MD, USA).

### 2.6. A Laser-Induced CNV Model and Measurement of CNV Volumes

A murine laser-induced CNV model was produced as previously described in our papers [15,16]. Briefly, the eyes of C57BL/6 mice were dilated by a combination of 0.5% tropicamide and 0.5% phenylephrine (Santen Pharmaceutical, Osaka, Japan) and the mice were anesthetized by a combination of midazolam (Sandoz, Tokyo, Japan), medetomidine (Orion, Espoo, Finland) and butorphanol tartrate (Meiji Seika Pharma, Tokyo, Japan), termed ‘MMB’. After anesthesia, the eyes of mice were covered with a contact lens (Haag-Streit Diagnostics, Koniz, Switzerland) to see the retinas clearly. A total of 5 laser spots (532 nm argon laser, 200 mw, 100 ms, 75 mm) were placed between the retinal vessels, located 2-disc diameters from the optic nerve head. During irradiation by laser, an air bubble was used as an index of Bruch’s membrane disruption. Laser lesions that lack the air bubble or have occurrence of hemorrhage were excluded from data analyses [15,16]. At day 7 after irradiation, the mice were anesthetized by MMB, followed by euthanasia, and the eyes were enucleated by forceps. The retinas from the eyes were flat-mounted and stained with isolectin B4 (Invitrogen, Carlsbad, CA, USA) [15,16]. We observed CNV with a laser microscope (Zeiss, Oberkochen, Germany) and measured a volume of CNV using Imaris (Bitplane, Zurich, Switzerland) as previously described in our papers [15,16].

### 2.7. A Light-Induced Retinopathy (LIR) Model and Measurement of Retinal Function by Electroretinography (ERG)

Development of a murine light-induced retinopathy (LIR) model was modified from our previous study [26], the exposure of 3000 lux white light to the eyes for 1 h. After 1 week following the light exposure, the eyes of BALB/c mice were dilated by a combination of 0.5% tropicamide and 0.5% phenylephrine (Santen Pharmaceutical, Osaka, Japan) and the mice were anesthetized by MMB under a dark room. After anesthesia, retinal function was evaluated by ERG, as previously described in our paper [18]. Briefly, active electrodes were recorded in the contact lens and a reference electrode was placed in the mouth. ERG responses were obtained from both eyes of each animal. Scotopic responses were recorded with various stimuli. The amplitudes of a-wave were measured from the baseline to the lowest point of a-wave. The amplitudes of b-wave were measured from the lowest peak of a-wave to the highest peak of b-wave. During the procedure, all mice were kept in warm conditions using heat pads.

### 2.8. Statistical Analysis

Data analyses were performed using Prism 5 (GraphPad, San Diego, CA, USA). Statistically significant differences were calculated using a two-tailed Student’s *t*-test or one-way ANOVA followed by a Bonferroni post hoc test. *p*-value of less than 0.05 was regarded as statistically significant.

## 3. Results

### 3.1. Rice Bran or Vitamin B6 Shows Inhibitory Effects on HIF Activation in ARPE-19 Cells under a CoCl_2_-Induced Hypoxic Condition

Food ingredients (with their expected components) were screened for inhibition of HIF activity (Figure 1A and Table A1). For the first gross screening, 661W cells (mouse immortalized cone photoreceptor cells with some features of retinal ganglion precursor-like cells) were utilized, as this cell line has been widely used as one of the in vitro cell models for ophthalmic drug development [27,28] and a HIF-luciferase reporter stable cell line has already been established [15,16]. Screened samples were in order of strong inhibitory tendencies on HIF activation (Table A1). At the first line, the top 10 samples with strong inhibitory tendencies were selected among 202 samples on HIF activation under a CoCl_2_-induced hypoxic condition (number 1–10 in Table A1). After collecting the 10 samples (garcinia fruit extract with or without water solubility and lactoferrin and lactoferrin from milk were considered as one sample after the gross first screening), we carefully evaluated inhibitory effects of them again at the second screening with the same system. We found that only six samples consistently showed statistically significant inhibitory effects on HIF activation under a CoCl_2_-induced hypoxic condition (Table A2).

We proceeded to the next final screening with these six samples. For the third final screening, we used ARPE-19 cells (human retinal pigmented epithelium cells) as this cell line also has been widely used for ophthalmic drug development [27,29]. In addition, this cell type, RPE cells, has been suggested as one of main pathological reasons for the development of CNV, finally leading to AMD [30,31,32]. Through the final screening, we found that six samples showed statistically significant HIF inhibitory effects (Figure 1B and Figure A1 and [15,16]). Taken together, four food ingredients (with their expected two component compounds, hydroxycitric acid and vitamin B6) were positively selected as inhibitors of HIF activation, as listed ‘*Garcinia cambogia*’, ‘lactoferrin’, ‘rice bran (*Oryza sativa* Linne, *Gramineae*, defatted)’ and ‘ginseng’ (Figure 1A). Based on the screening results, we could demonstrate therapeutic effects of lactoferrin and *Garcinia cambogia* (and its abundant component hydroxycitric acid) via inhibition of HIF activation in murine models of CNV [15,16].

Next, with the rest of the positively selected food ingredients (rice bran or ginseng) from the screenings, we further attempted to examine which components inside defatted rice bran or ginseng could help it to exert HIF inhibitory effects. While we could not find which components inside ginseng could help it to have HIF inhibitory effects, among the components contained in rice bran (Table A3), we have found that vitamin B6 showed a significant and the most robust HIF inhibitory effect (Figure 1B and Figure A2). Taken together, in this current study, we mainly focused on unraveling potent effects of rice bran and vitamin B6 as novel HIF inhibitors. For further experiments under a CoCl_2_-induced hypoxic condition, we examined whether rice bran or vitamin B6 has cellular toxicity using MTT assay (Figure A3). Basically, cytotoxicity of them was not significantly detected although there was a decreasing tendency in mitochondrial activity in high-dose vitamin B6 (1 mg/mL)-treated group.

### 3.2. Rice Bran or Vitamin B6 Has Suppressive Effects on HIF Stabilization in ARPE-19 Cells under a CoCl2-Induced Hypoxic Condition

Suppressive effects of rice bran and vitamin B6 on HIF stabilization in protein levels were examined (Figure 2). HIF-1α expression was stabilized in ARPE-19 cells 6 h after incubation of 200 µM of CoCl_2_. Then, stabilized HIF-1α expression was significantly reduced by rice bran and vitamin B6 treatments, respectively. On the other hand, in 661W cells, there was no statistical difference by rice bran or vitamin B6 treatment in stabilized HIF-1α expression 6 h after incubation of 200 µM of CoCl_2_, (Figure A4). These results indicate that rice bran and vitamin B6 could have suppressive effects on HIF-1α stabilization in RPE cells more than neuronal cells.

Next, we examined whether rice bran and vitamin B6 could act on another HIF expression (HIF-2α) in ARPE-19 cells under the same condition (Figure A5). We could not see a significant increase in HIF-2α expression under a CoCl_2_-induced hypoxic condition, and rice bran and vitamin B6 did not change its expression. Taken together, it indicates that HIF-1α (rather than HIF-2α) might be the major regulator in ARPE-19 cells under this hypoxic condition, which is in agreement with several previous reports using ARPE-19 cells [33,34,35].

### 3.3. Rice Bran or Vitamin B6 Inhibits the HIF/VEGF Axis in ARPE-19 Cells under a CoCl2-Induced Hypoxic Condition

Suppression on HIF-1α stabilization in RPE cells could be followed by inhibition of DNA binding of stabilized HIF-1α, finally reducing induction of HIF target gene expressions, especially VEGF. Thus, we examined whether rice bran or vitamin B6 suppresses HIF downstream-target hypoxia-responsive gene expressions under a CoCl_2_-induced hypoxic condition (Figure 3). A total of 8 h after 200 µM of CoCl_2_ incubation in ARPE-19 cells, downregulation of *HIF-1*α mRNA expression was detected due to a negative feedback from the post translational HIF-1α protein modification, which has been consistently seen in several previous papers [15,16,19,36]. CoCl_2_ induced upregulation of *VEGF*, *BNIP3* and *PDK1* mRNA expressions as results of HIF activation. Upregulated *VEGF* mRNA expression was significantly reduced by rice bran treatment. Although upregulated *VEGF* mRNA expression was not significantly reduced by vitamin B6 treatment, its expression showed a decreasing tendency. On the other hand, upregulated *BNIP3* and *PDK1* mRNA expressions were not significantly reduced by rice bran treatment although these expressions were significantly reduced by vitamin B6 treatment.

### 3.4. Rice Bran or Vitamin B6 Administration Suppresses Retinal Neovascularization in a Murine Model of CNV

To assess therapeutic effects of rice bran and vitamin B6 on pathological neovascularization, a vitamin B6-deficient diet (AIN-93G, Oriental Yeast Co. Ltd., Tokyo, Japan) supplemented with vitamin B6 (9 or 35 mg/kg diet weight, DSM Nutritional Products Ltd., Kaiseraugst, Switzerland) or rice bran (defatted, 587.5 mg/kg diet weight, Oryza Oil & Fat Chemical Co., Ltd., Ichinomiya, Japan) with vitamin B6 (1 mg/kg diet weight) was given to 4-weeks-old male mice for total 7 weeks (Figure 4A). The control group was provided with the basic diet (AIN-93G) which only contains 1 mg of vitamin B6 in 1 kg of the diet. This amount of vitamin B6 is lower than that of normal diet commonly used for mouse studies; about 5–8 mg of vitamin B6 in 1 kg of diet, of which value could be considered as the roughly recommended amount for a daily life consumption.

After 3 days of irradiation by laser which was performed on week 6 after administration of diet supplemented with rice bran or vitamin B6, we could see stabilized HIF-1α expression in the laser-irradiated retina and its expression was reduced by the rice bran- or vitamin B6-administered retina (Figure 4B). After 1 week of irradiation by laser, we quantified CNV volume as previously described [15,16]. A decreased volume of CNV was significantly observed in the rice bran- or vitamin B6-administered mice in comparison with that in the control mice, respectively (Figure 4C).

### 3.5. Rice Bran or Vitamin B6 Administration Did Not Directly Affect Neuronal Dysfunction in a Murine Model of LIR

To assess the therapeutic effects of the dietary supplement on protection in retinal function, we employed a murine LIR model of retinal degeneration. In this model, we could observe retinal dysfunction 1 week after the light exposure analyzed by ERG (Figure A6), which is in the basic agreement with previous observations [26,37,38].

Next, we attempted to examine therapeutic effects of rice bran and vitamin B6 on direct retinal protection. A diet supplemented with rice bran or vitamin B6 was given to 5-weeks-old male mice for a total 8 weeks (Figure 5A). The control group was provided with a normal diet (which basically contains 1 mg/kg diet weight of vitamin B6) without the additional supplement. A total of 1 week after the light exposure which was performed on week 7 after administration of the diet supplemented with rice bran or vitamin B6, we found no significant change in the amplitudes of a- and b-waves among all of the groups except for the high-dose vitamin B6-administered group (Figure 5B,C).

## 4. Discussion

Rice bran is a hard-outer covering of rice grains and it is produced as a byproduct of milling for the production of refined rice [39]. In East Asia, people simply use rice bran to enrich some dishes and wish to increase their intake of dietary fiber as it contains vitamins, minerals, fatty acids, dietary fiber, and other sterols [39]. Rice bran has been traditionally thought to be a normal fiber material and considered enough for making cooking oil or feeding their livestock such as cattle or pigs. However, its high nutritional values received considerable attention after their therapeutic potentials were reported in various diseases [40]. In the current study, we found a new aspect of uses of rice bran as an HIF inhibitor and suggested vitamin B6 would be a working compound for this role among the components.

HIF plays a critical role in the maintenance of cellular homeostasis responded by alteration of oxygen status [8,9,10]. However, under pathological hypoxic conditions, HIF activation can cause devastating outcomes, in this case, retinal neovascularization [11]. Previously, we demonstrated that pharmacological intervention via administration of doxorubicin or topotecan [18], marine products [17], and a mushroom product [19] showed therapeutic effects on retinal neovascularization through inhibition of HIF activation and suppression of its downstream VEGF expression. In addition, we also demonstrated a significant reduction of CNV volume in RPE-specific *Hif-1*α-conditional knockout mice in comparison with that in control mice [15,41], which implies that genetic intervention of HIF is also a beneficial target against pathological hypoxic conditions in the eyes. In our current study, oral administration of rice bran or vitamin B6 as a diet supplement also showed similar therapeutic effects on retinal neovascularization. Taken together, we think that inhibition of HIF activation could have benefits for managing pathological ocular neovascularization.

The development of AMD involves complex pathological mechanisms. Above all, increasing evidence suggests that dysfunction of RPE is crucially involved in neovascular and atrophic forms of AMD [42]. RPE is a monolayer of polarized cells and lies underneath the retina intimately interacting with photoreceptor cells, of which communication is important for retinal homeostasis [43]. RPE controls the outer blood-retinal barrier via regulation of nutrient and oxygen delivery to the retina and clearance of metabolic waste from photoreceptor cells. RPE physiologically produces growth factors such as VEGF to support the retina and choriocapillaris [42,44,45]. Damages could occur to RPE over many years with ageing and increasing pathological stresses, which causes dysfunction of RPE and following pathological release of growth factors from RPE, especially VEGF [30,31]. This may be one of main pathological reasons for the development of CNV that finally leads to AMD [32]. In our study, upregulated *VEGF* mRNA expression in APRE-19 cells under a CoCl_2_-induced hypoxic condition was reduced by rice bran treatment. However, more data are required to unravel the in vivo mode of action regarding the reduction in VEGF levels in the eye by administration of rice bran or vitamin B6. This will be further studied. Moreover, CNV is one of the complex tissues in the eye composed of vascular components including endothelial cells, vascular smooth muscle cells, and pericytes, and extravascular cells such as inflammatory cells, fibroblasts, glial cells, and RPE cells [46,47,48]. Recently, endothelial cells, vascular smooth muscle cells, macrophages, and RPE cells have been suggested to be main cell types of CNV formation [46,47,48]. It is still hard to tell which cells exactly give the most contribution of CNV formation. Therefore, we think that more comprehensive studies regarding the relationship of VEGF-producing cell types with contribution of CNV formation may be needed for a better understanding of pathological mechanisms for ocular neovascularization. Nonetheless, based on what we have found so far, we suggest that a strategy to control RPE using rice bran or vitamin B6 might be applicable for AMD therapy.

Secondly, we examined a direct therapeutic role of rice bran (along with vitamin B6) in the retina. However, we could not see the positive effect. Instead, high-dose vitamin B6 administration caused retinal damages, which implies an excessive amount of vitamin B6 may not be suitable for retinal function. Furthermore, stabilized HIF-1α expression in neuronal cells under a CoCl_2_-induced hypoxic condition was less affected by rice bran and vitamin B6. Although more studies are required, rice bran may directly work on RPE cells under pathological conditions in the eye rather than to work on neuronal cells.

Hypoxia-responsive genes other than *VEGF* were also examined in our study. Interestingly, *BNIP3* and *PDK1* expressions were not dramatically reduced by rice bran treatment while *VEGF* expression showed the expected result. Furthermore, *BNIP3* and *PDK1* expressions were significantly reduced by vitamin B6 treatment. This might be explained by the fact that rice bran contains several components other than vitamin B6 [39], which could induce other unknown signaling pathways to hinder inhibition of hypoxia-responsive gene induction under a CoCl_2_-induced hypoxic condition. More comprehensive studies regarding unknown effects of each component in rice bran may be further needed for a better understanding of pathological mechanisms for ocular neovascularization.

To date, effective treatments for pathological retinal neovascularization contain intravitreal injection of anti-VEGF drugs as well as laser photocoagulation, topical injection of corticosteroids, and vitreoretinal surgery [4,49].Chronic anti-VEGF therapies may induce photoreceptor cell atrophy [50] as it could abrupt normal physiological roles of VEGF for retinal function [51]. In addition, these treatments are very of high cost and even invasive to the eye, which is not patient-friendly [52,53]. In our current study, we targeted pathological HIF activation using rice bran to reduce HIF-induced pathological *VEGF* expression. In addition, edible rice bran is cost-effective and noninvasive to patients as it is always produced as a byproduct of milling for the production of rice [39] and possesses its beneficial features of good patient compliance and fear or pain avoidance [54].

## 5. Conclusions

In conclusion, we screened various types of food ingredients as prospective HIF inhibitors and demonstrated that rice bran and its component vitamin B6 possess HIF inhibitory effects. Furthermore, they showed their suppressive effects on pathological retinal neovascularization in a murine CNV model. Rice bran or vitamin B6 dietary supplement could be useful as stand-alone therapies or as adjuvants to anti-VEGF therapies for neovascularization in ocular diseases.

## 6. Patents

The data of the current research have applied for a patent (application number: 2019-038043).

## Figures and Tables

**Figure 1 ijms-21-08940-f001:**
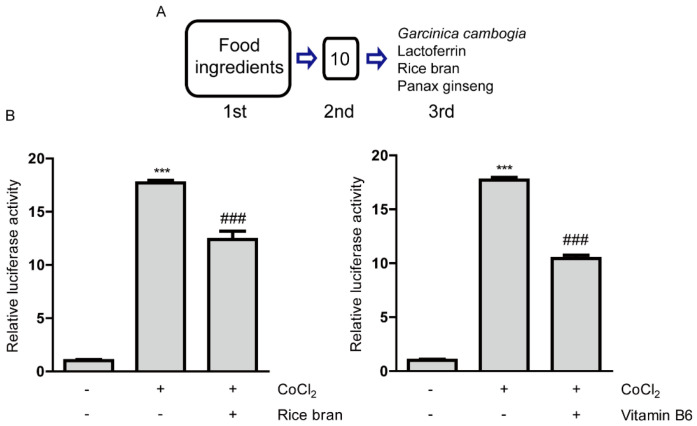
Inhibitory effects of rice bran and vitamin B6 on hypoxia-inducible factor (HIF) activity. (**A**) A process of drug screenings for HIF inhibitors. After the first screening, 10 samples were shown to be positive. After the second screening, four food ingredients (lactoferrin, rice bran, panax ginseng and *Garcinia cambogia*) with their expected 2 component compounds (hydroxycitric acid and vitamin B6) were selected as HIF inhibitors. (**B**) Quantitative analyses of HIF-reporter luciferase assay using ARPE-19 cells (*n* = 3 per group) showed that rice bran (1 mg/mL) and vitamin B6 (1 mg/mL) inhibited HIF activity induced by 200 µM CoCl_2_. *** *p* < 0.001, ### *p* < 0.001, compared with no treatment and 200 µM of CoCl_2_ treatment, respectively. Bar graphs were presented as mean with the ± standard deviation. The data were analyzed using one-way ANOVA followed by a Bonferroni post hoc test. Solvents, rice bran: DMSO; vitamin B6: water.

**Figure 2 ijms-21-08940-f002:**
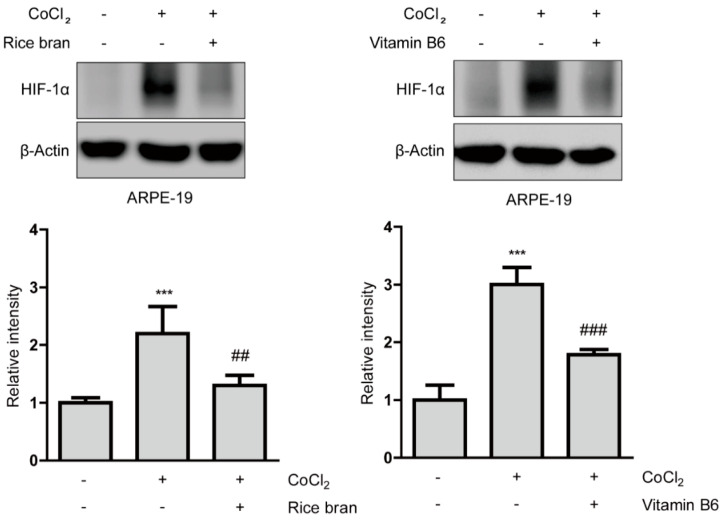
Suppressive effects of rice bran and vitamin B6 on HIF-1α stabilization. Representative immunoblot images and quantitative analyses (*n* = 4 per group) for HIF-1α and β-Actin showed that HIF-1α was stabilized in ARPE-19 cells under a CoCl_2_-induced pseudo-hypoxic condition. Rice bran (1 mg/mL) and vitamin B6 (1 mg/mL) significantly decreased stabilized HIF-1α expression. *** *p* < 0.001, compared with no treatment, ## *p* < 0.01, ### *p* < 0.001, compared with CoCl_2_ treatment. Bar graphs were presented as mean ± standard deviation. The data were analyzed using one-way ANOVA followed by a Bonferroni post hoc test. Solvents, rice bran: DMSO; vitamin B6: water.

**Figure 3 ijms-21-08940-f003:**
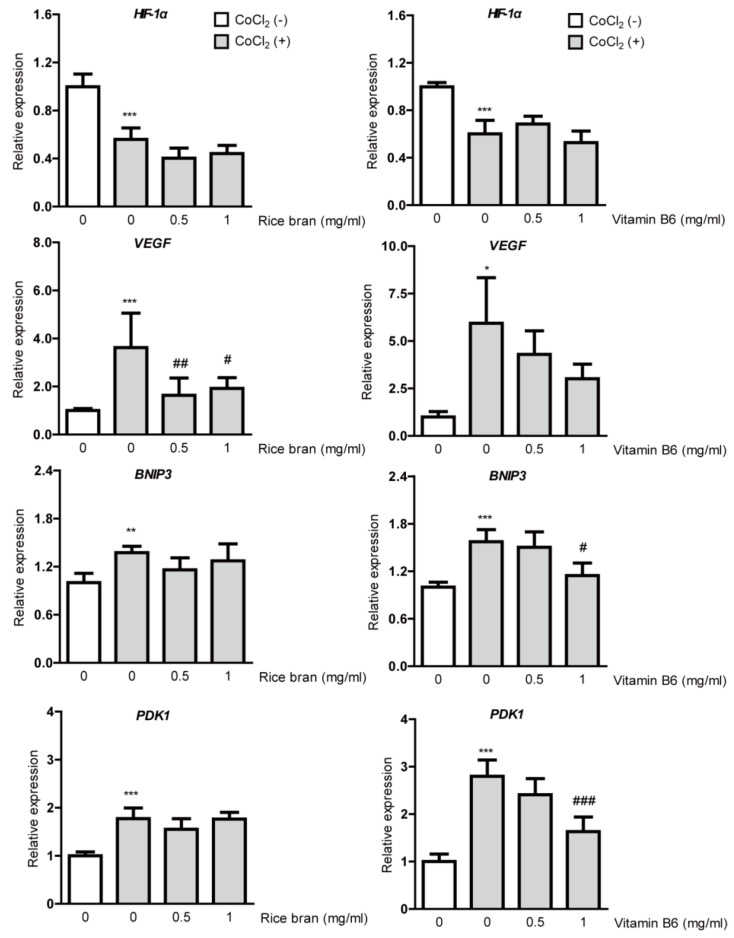
Suppression of hypoxia-responsive gene expressions by rice bran and vitamin B6. Quantitative analyses (*n* = 4–6 per group) showed significant changes in *HIF-1α*, *VEGF*, *BNIP3* and *PDK1* mRNA expressions 8 h after incubation of CoCl_2_ in ARPE-19 cells. Upregulated *VEGF* mRNA expression was significantly reduced by rice bran treatment. There was a decreasing tendency of upregulated *VEGF* mRNA expression by vitamin B6 treatment. * *p* < 0.05, ** *p* < 0.01, *** *p* < 0.001, compared with no treatment, # *p* < 0.05, ## *p* < 0.01, ### *p* < 0.001, compared with CoCl_2_ treatment. Bar graphs were presented as mean with ± standard deviation. The data were analyzed using one-way ANOVA followed by a Bonferroni post hoc test. Solvents, rice bran: DMSO; vitamin B6: water.

**Figure 4 ijms-21-08940-f004:**
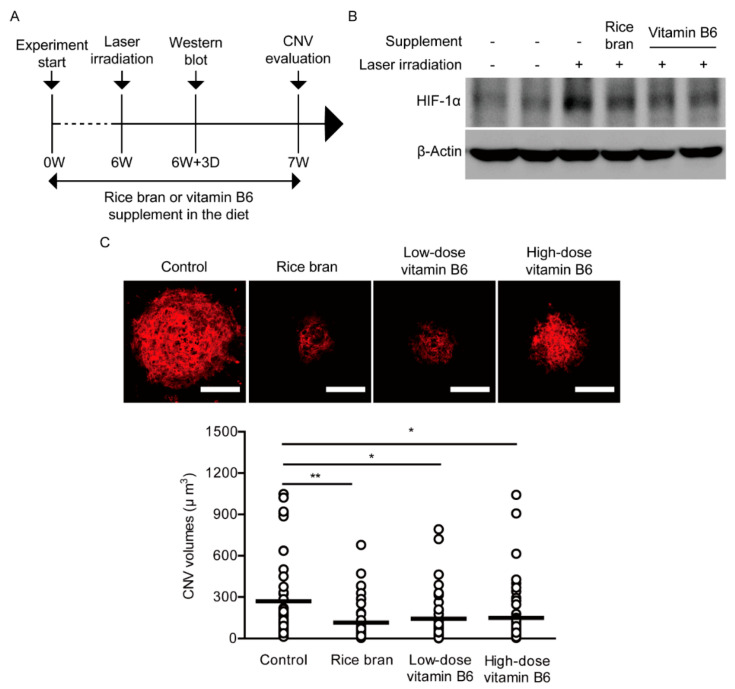
Suppression of neovascularization by rice bran and vitamin B6. (**A**) A schematic illustration demonstrates the murine choroidal neovascularization (CNV) model procedure and administration of rice bran or vitamin B6 to mice. (**B**) An immunoblot image for HIF-1α and β-Actin in the retina with or without the supplement of rice bran or vitamin B6, 3 days after the laser irradiation. (**C**) Representative images of CNV in the whole mount staining with isolectin B4 and quantitative analyses (*n* = 5–6 per group, *n* = 42–49 laser spots in the eyes per group) showed that the volume of CNV was significantly reduced by administration of rice bran (587.5 mg/kg diet weight) and vitamin B6 (9 or 35 mg/kg diet weight), respectively. Scale bars, 100 μm. * *p* < 0.05, ** *p* < 0.01. Dot plot graphs were presented as mean. The data were analyzed using Student’s *t*-test.

**Figure 5 ijms-21-08940-f005:**
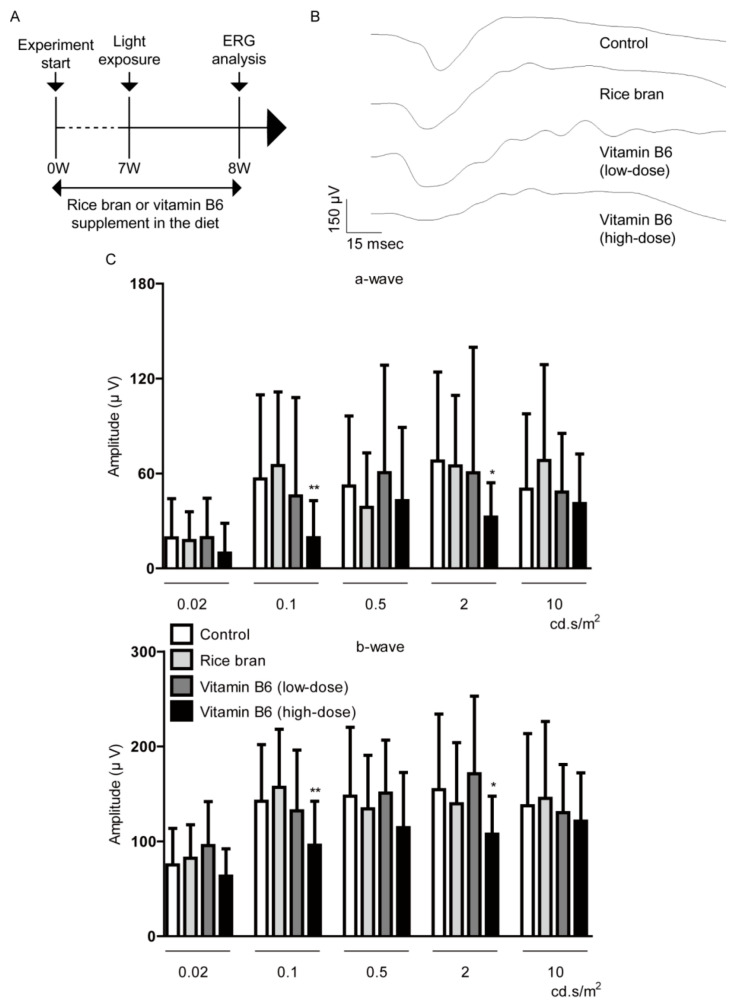
Direct retinal protection by rice bran and vitamin B6. (**A**) A schematic illustration demonstrates the murine light-induced retinopathy (LIR) model procedure and administration of rice bran or vitamin B6 to mice. (**B**,**C**) Representative waveforms of a- and b-waves (2 cd.s/m^2^) and quantitative analyses showed that rice bran (587.5 mg/kg diet weight) or vitamin B6 (9 mg/kg diet weight) did not change the amplitudes of a-wave and b-wave in LIR mice (*n* = 9–10 per group, 18–20 eyeballs per group). There was a significant decrease in the amplitudes of a-wave and b-wave in high-dose vitamin B6 (35 mg/kg diet weight)-administered LIR mice. * *p* < 0.05, ** *p* < 0.01, compared with control. The data were analyzed using Student’s *t*-test.

**Table 1 ijms-21-08940-t001:** Primer list.

Name	Direction	Sequence (5′ → 3′)
*GAPDH*	Forward	TCCCTGAGCTGAACGGGAAG
	Reverse	GGAGGAGTGGGTGTCGCTGT
*HIF-1α*	Forward	TTCACCTGAGCCTAATAGTCC
	Reverse	CAAGTCTAAATCTGTGTCCTG
*VEGF*	Forward	TCTACCTCCACCATGCCAAGT
	Reverse	GATGATTCTGCCCTCCTCCTT
*BNIP3*	Forward	GGACAGAGTAGTTCCAGAGGCAGTTC
	Reverse	GGTGTGCATTTCCACATCAAACAT
*PDK1*	Forward	ACAAGGAGAGCTTCGGGGTGGATC
	Reverse	CCACGTCGCAGTTTGGATTTATGC

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
