# Peer review of "Rice Bran and Vitamin B6 Suppress Pathological Neovascularization in a Murine Model of Age-Related Macular Degeneration as Novel HIF Inhibitors"

_ijms, 2020, doi:10.3390/ijms21238940_

Round 1

Reviewer 1 Report

The manuscript by Ibuki and colleagues demonstrated that rice bran and vitamin B6 are novel effective HIF inhibitors identified by in vitro screening assays in two different cell lines. The authors further demonstrated that rice bran and lower-dose vitamin B6 can effectively suppress choroidal neovascularization. The manuscript is well-written and the experimental design is appropriate. Results from this manuscript suggest potential clinical benefits for patients with wet-form age-related macular degeneration. It is recommended to be accepted after comments are addressed:

Major concern:

  • What is the effect of rice bran and vitamin B6 on cell survival? Is the suppressive effect on HIF activity due to cellular toxicity?

Minor concerns:

  1. Info of the dose of Rice bran and Vitamin B6 in Fig.1 and 2 is missing. Please add the info in the figure legends. 
  2. Lines 32, 212 to 218, 307, 317-319: gene name should be all capitals as the results from the human cell line (i.e.) VEGF, BNIP3, etc.

Reviewer 2 Report

Mari Ibuki and coauthors screened 202 compounds for prospective HIF inhibition utilizing 2 cell lines (ARPE-19 and 661W). Within these they identified 2 promising candidates which are Rice Bran and a component of it Vitamin B6. These compounds showed an inhibitory effect on HIF activation and suppressed Vegf mRNA upregulation under CoCl2-induced pseudo-hypoxic condition. In addition, a murine AMD model was utilized for examining suppressive effects of the ingredients on retinal neovascularization. As a result, dietary supplement of the compounds significantly suppressed retinal neovascularization in the AMD model.

Overall this manuscript is well written and the data are presented clearly.

Yet one major concern lies on the fact that the authors claim that the in vivo suppressing effect of Rice Bran and Vitamin B6 on retinal neovascularization in the AMD model might be explained by inhibition of VEGF in the retina. This needs to be shown.

In addition, there are a few more concerns, addressing which may help improve the rigor of this study.

Major concerns:

  1. The authors should explain, why they saw in the first screen in the mouse cone photoreceptor 661W cell line an effect of Rice Bran and Vitamin B6, which was then lost (Figure A3)?
  2. It should be also mentioned/discussed; which other cells (microglia, EC….?) are known to produce relevant VEGF levels in the retina? Are there differences in the effect on neovascularization depending which cell type is targeted for Hif?
  3. The authors studied the effect of an inhibitory effect on HIF-1α activation. What about HIF-2α? Was is also regulated by the compounds? There are several in vivo studies showing effects of Hif2α deficiency on pathological retinal Neovascularization. (e.g.: global knockdown of Hif2α: PMID: 12606578; astrocyte‐specific: PMID: 20544853;hematopoietic-specific Hif2α deficiency; PMID: 30156910……).
  1. Line 236:The control group was provided with the basic diet (AIN-93G) which only contains 1 mg of vitamin B6 in 1 kg of the diet. Please state whether this is a normal physiological concentration of vitamin B6 for mice…. or is it lower, then in normal food.
  1. Figure 4: Representative 249 images of CNV in the whole mount staining with isolectin B4 and quantitative analyses (n = 5-6 per group)…. In the graph I see more dots/ data points, even when I consider n=mouse, and each mouse has 32 eyes there are more dots…. Please explain
  2. Although Rice Bran and Vitamin B6 showed VEGF suppressive effects in vitro on mRNA level…. What about protein levels? In addition, the changes of VEGF expression level in vivo need to be investigated. Were retinal levels of VEGF and/ or other hypoxia regulated angiogenic factors (e.g. erythropoietin) changed in response to Rice Bran and/or Vitamin B6 treatment in the AMD model?
  3. Do Rice Bran and Vitamin B6 affect physiological angiogenesis? Are there data or are there other known side effects of the compounds, which could limit their usage?

Minor concerns:

  1. In the Materials and Methods part in 2.2, line 99.: cells were co-treated with 1 mg/ml of each ingredient….How were the substances diluted/ in which medium/ are they all water soluble?/ what are the controls?
  1. In the Materials and Methods part in 2.5., line 123: mice were anesthetized by MMB…. and the eyes were enucleated by forceps…When did you kill the mice?
  1. In the Materials and Methods part in 2.7., line 143: Please specify which student’s t-test was used. 1- or 2 tailed?
  1. In Figure 2: please include the concentration of the compounds in the text
  2. In Figure 2 and 3: Rice bran and vitamin B6 significantly decreased stabilized HIF-1α expression. ***P < 0.001, ##P < 0.01, ###P < 0.001 compared with no treatment and CoCl2 treatment, respectively….. This is not correct…. only compared to CoCl2 treatment
  1. Figure 4: the size of the scale bar is missing…..
  2. Fig A1, line 364: Ginseng dramatically inhibited……. Please delete dramatically…. It is only a very mild effect
